# Influence of Butanol Isomerization on Photothermal Hydrogen Production over Ti@TiO₂ Core-Shell Nanoparticles

**Sara El Hakim, Mathéo Bathias, Tony Chave and Sergey I. Nikitenko \***

Institute for the Separation Chemistry in Marcoule (ICSM), University Montpellier, UMR 5257, CEA, CNRS, ENSCM, Marcoule, F-30207 Bagnols sur Cèze, France
**\*** Correspondence: serguei.nikitenko@cea.fr

**Abstract:** In this work, we reported for the first time the effect of butanol isomerization on the photothermal production of hydrogen in the presence of a noble, metal-free Ti@TiO₂ core-shell photocatalyst. The experiments were performed in aqueous solutions of 1-BuOH, 2-BuOH, and t-BuOH under Xe lamp irradiation (vis/NIR: 8.4 W, UV: 0.6 W) at 35–69 °C. The increase in temperature significantly enhanced $H_2$ formation, indicating a strong photothermal effect in the studied systems. However, in dark conditions, $H_2$ emission was not observed even at elevated temperatures, which clearly points out the photonic origin of $H_2$ photothermal formation. The rate of $H_2$ production followed the order of 1-BuOH >> 2-BuOH > t-BuOH in the entire range of studied temperatures. In the systems with 1-BuOH and 2-BuOH, hydrogen was the only gaseous product measured online in the outlet carrier argon using mass spectrometry. By contrast, a mixture of $H_2$, $CH_4$, and $C_2H_6$ was detected for t-BuOH, indicating a C–C bond scission with this isomer during photocatalytic degradation. The apparent activation energies, $E_a$, with 1-BuOH/2-BuOH isomers (20–21 kJ·mol$^{-1}$) was found to be larger than for t-BuOH (13 kJ·mol$^{-1}$). The significant difference in thermal response for 1-BuOH/2-BuOH and t-BuOH isomers was ascribed to the difference in the photocatalytic mechanisms of these species. The photothermal effect with 1-BuOH/2-BuOH isomers can be explained by the thermally induced transfer of photogenerated, shallowly trapped electron holes to highly reactive free holes at the surface of TiO₂ and the further hole-mediated cleavage of the O-H bond. In the system with t-BuOH, another mechanism could also contribute to the overall process through hydrogen abstraction from the C–H bond by an intermediate •OH radical, leading to CH₃• group ejection. Formation of •OH radicals during light irradiation of Ti@TiO₂ nanoparticle suspension in water has been confirmed using terephthalate dosimetry. This analysis also revealed a positive temperature response of •OH radical formation.

**Keywords:** photocatalysis; hydrogen production; butanol; isomers; photothermal effect; titanium

## 1. Introduction

Environmentally friendly industry is likely to be a dominant end-use for hydrogen as the world strives to achieve net zero by 2050 [1,2]. Catalytic processes play a key role in the production of low-carbon hydrogen [3,4]. More specifically, photocatalytic hydrogen evolution from bio-based sources, such as alcohols, has a great potential for green hydrogen production and can also be used effectively for the production of value-added aldehydes, carboxylic acids, and esters [5]. Note that, although $CO_2$ is also released in these processes, it is considered to be a part of a clean cycle when the sacrificial reagent has a biological origin [6]. Introducing heat into photocatalytic processes has attracted much attention in the last decade because it may significantly improve photoconversion efficiency [7,8]. However, the data about dominating photothermal mechanisms during $H_2$ production are still scarce in the literature. Recent studies of the H/D kinetic isotope effect revealed that the photothermal effect in the process of reforming aqueous glycerol over a Ti@TiO₂ core-shell photocatalyst is mainly attributed to the electron-hole-mediated splitting of the

O-H bond from glycerol [9]. On the other hand, the dynamics of intermediates at the catalyst surface could also contribute to the overall reaction kinetics through the entropy of activation. It was concluded that this effect reduces the influence of temperature on photocatalytic hydrogen production. Herein, we focus on photothermal $H_2$ production from aqueous solutions of butanol isomers in the presence of Ti@TiO$_2$ nanoparticles (NPs). Numerous studies of photocatalytic alcohol reforming over bare TiO$_2$ and noble metal-loaded TiO$_2$ NPs in aqueous solutions, and in a gas phase, revealed the involvement of $\alpha$-H atom abstraction in the limiting stage for primary and secondary alcohol isomers, followed by the formation of aldehydes or ketones, respectively [10–13]. On the other hand, the photoreforming of t-BuOH in the gas phase over TiO$_2$ yields butyraldehyde as a primary product [14,15]. In aqueous solutions, photocatalytic degradation of t-BuOH over platinized TiO$_2$ is accompanied by C–C bond scission and formation of an $H_2$, $CH_4$, $C_2H_6$ gas mixture [16]. To the best of our knowledge, the influence of butanol isomerization on photothermal $H_2$ production in aqueous solutions has not been reported in the literature. Meanwhile, a comparative study of butanol isomers would provide new insights onto the photothermal reaction mechanism.

## 2. Results and Discussion

The experiments were performed using Ti@TiO$_2$ core-shell NPs obtained by sonohydrothermal treatment (SHT) of metallic titanium Ti$^0$ NPs in pure water. The experimental details are described in Materials and Methods. HR TEM images (Figure 1) show the quasi-spherical air-passivated Ti$^0$ NPs (50–150 nm) before SHT treatment. These particles tend to exhibit a nanocrystalline shell composed of 10–20 nm anatase TiO$_2$ crystals after SHT treatment. Detailed analysis of Ti@TiO$_2$ NPs has been reported in our earlier studies [17–19]. It was shown that Ti@TiO$_2$ NPs provide an extended photo response in the UV/vis/NIR spectral range due to the interband/intraband transitions of the metallic core and the bandgap of the semiconducting titanium oxide shell ($3.52 \pm 0.01$ eV [18]), thus allowing efficient solar light harvesting. Photocatalytic activity of this material was ascribed to the effective electron-hole separation between the Ti$^0$ core and the nanocrystalline TiO$_2$ shell. This conclusion is supported by the fact that the non-coated Ti$^0$ NPs do not exhibit photocatalytic activity despite strong light absorption [17,18].

### (A) Ti$^0$ NPs     (B) Ti@TiO$_2$ SHT > 150 °C

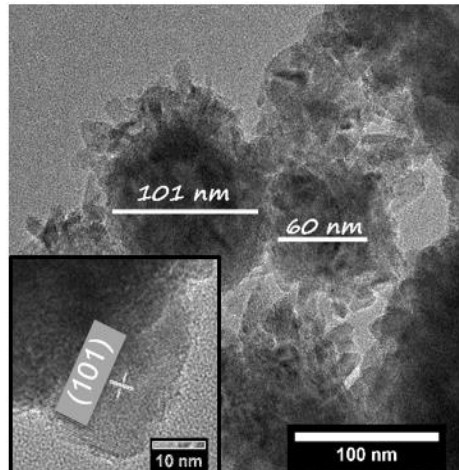

**Figure 1.** Typical HR TEM images of Ti$^0$ (**A**) and Ti@TiO$_2$ (**B**) NPs. In the inset, the distance of 0.35 nm corresponds to (101) plane of TiO$_2$ anatase. The analysis was performed with a Jeol 2200FS (200 kV) device.

Figure 2 depicts a typical profile of $H_2$ emission obtained using a 0.5 M 1-BuOH aqueous solution irradiated using an Xe lamp in the presence of suspended Ti@TiO$_2$ NPs. The 2-BuOH and t-BuOH solutions exhibit similar behavior. A stepwise increase in the bulk temperature causes a significant stepwise increase in $H_2$ concentration in the outlet gas, indicating a strong photothermal effect in the studied system. However, when the light is cut off, $H_2$ formation is not observed even at elevated temperatures, which clearly indicates the photonic origin of $H_2$ formation. It should be noted that the strong evaporation of BuOH isomers does not allow us to study $H_2$ photothermal production at T > 69 °C.

In the entire range of the studied temperatures, the $H_2$ formation rate fell in the sequence of 1-BuOH >> 2-BuOH > t-BuOH as shown in Figure 3. This trend can be explained by the increased adsorption strength of primary alcohols on the metal oxide surface as compared to secondary and tertiary isomers [20]. It is worth noting that the isomers of alcohols exhibit such tendencies with many other TiO$_2$-based photocatalysts loaded with noble metals, M/TiO$_2$, where M = Pt, Pd, Au, Rh [21–23]. Table 1 summarizes the values of the apparent activation energy, $E_a$, calculated from Arrhenius plots obtained in the temperature range of 35–69 °C for all studied systems. Thus, it can be noticed that, despite the significant difference in reaction rate, both systems with 1-BuOH and 2-BuOH are characterized by very similar values of $E_a$. In addition, these values are comparable with the $E_a$ measured previously for reforming glycerol with the same catalyst [19], which implies a similarity in the reaction mechanism for these sacrificial reagents. Earlier studies have noticed that the photothermal effect during the photolysis of glycerol with Ti@TiO$_2$ NPs is attributed to the thermally induced transfer of photogenerated, shallowly trapped electron holes to highly reactive free holes, $h^+$, at the surface of TiO$_2$ [9]. Further electron-hole-mediated cleavage of the O-H bond from glycerol could lead to $H_2$ formation. Similarly, the photothermal $H_2$ production on Ti@TiO$_2$ in the presence of 1-BuOH/2-BuOH isomers can be expressed by the Equations (1)–(4):

Scavenger adsorption:

$$\text{Ti@TiO}_2 + \text{RCH}_2\text{OH}/\text{R}_1\text{R}_2\text{CHOH} \rightleftarrows \text{Ti@TiO}_2 \cdot \text{RCH}_2\text{OH}/\text{R}_1\text{R}_2\text{CHOH} \tag{1}$$

Photoexcitation:

$$\text{Ti@TiO}_2 \cdot \text{RCH}_2\text{OH}/\text{R}_1\text{R}_2\text{CHOH} + h\nu \rightarrow [^{e^-}\text{Ti@TiO}_2{}^{h+}] \cdot \text{RCH}_2\text{OH}/\text{R}_1\text{R}_2\text{CHOH} \tag{2}$$

Hole-mediated oxidation:

$$\text{RCH}_2\text{OH}/\text{R}_1\text{R}_2\text{CHOH} + 2h^+ \rightarrow \text{RCHO}/\text{R}_1\text{R}_2\text{CO} + 2\text{H}^+ \tag{3}$$

Molecular hydrogen formation:

$$2\text{H}^+ + 2e^- \rightarrow \text{H}_2 \tag{4}$$

where RCH$_2$OH and R$_1$R$_2$CHOH correspond to 1-BuOH and 2-BuOH, respectively. After light absorption by the catalyst, the adsorbed RCH$_2$OH and/or R$_1$R$_2$CHOH species are oxidized by photogenerated holes to form aldehydes and/or ketones and two protons. Molecular hydrogen is then obtained upon the reduction of produced H$^+$ cations by the photogenerated electrons. It is interesting to note that the photolysis of a 1-BuOH aqueous solution in the presence of TiO$_2$ loaded with noble metals leads to the formation of $H_2$, $CO_2$, $C_3H_8$ while that of 2-BuOH leads to $H_2$, $CO_2$, $C_2H_6$, $CH_4$ in the gas phase, respectively [10,24]. In our case, $H_2$ was the only gaseous photocatalytic product observed for both isomers. The absence of $CO_2$ emissions could be related to the low adsorption of the formed aldehydes/ketones on the surface of our catalyst, thus inhibiting their further oxidation into $CO_2$. This conclusion is supported by the absence of solution acidification during photolysis, indicating that organic acids are not formed during the process. However, for the system with t-BuOH, the formation of $CH_4$ and $C_2H_6$ was detected at 35–52 °C, along with $H_2$ formation, which agrees with the observation of

CH$_3$$^\bullet$ radicals during photolysis of t-BuOH in the gas phase over TiO$_2$ [14]. At higher temperatures, strong evaporation of t-BuOH did not allow for the proper measurements of hydrocarbons using mass spectrometry.

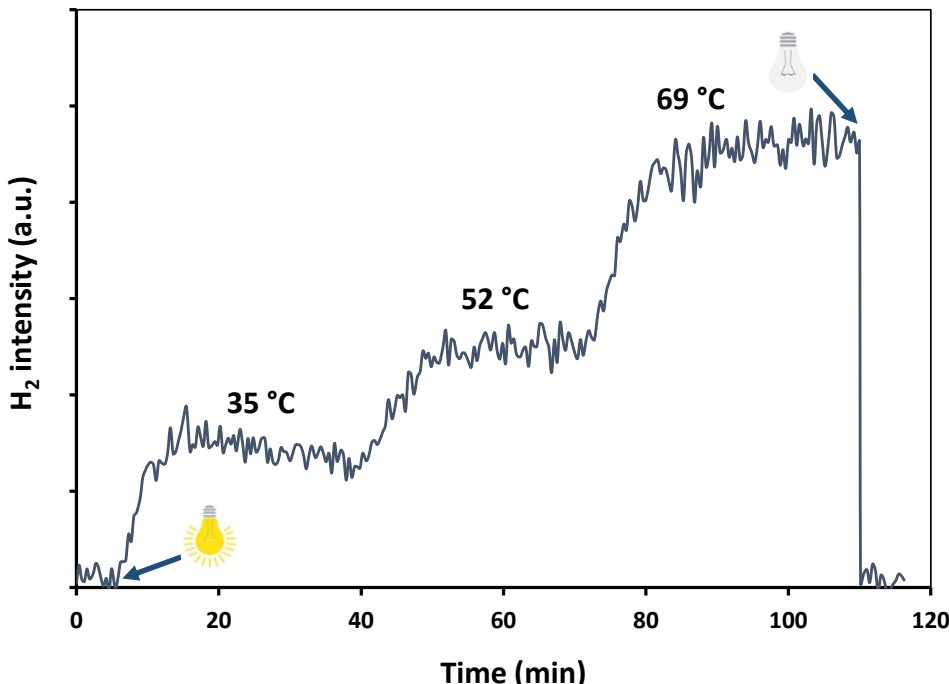

**Figure 2.** Typical temperature-dependent profile of H$_2$ emission during photolysis of 0.5 M 1-BuOH with white light of an Xe lamp under Ar flow (45 mL·min$^{-1}$) in the presence of Ti@TiO$_2$ NPs. H$_2$ was measured online in the outlet gas using mass spectrometry.

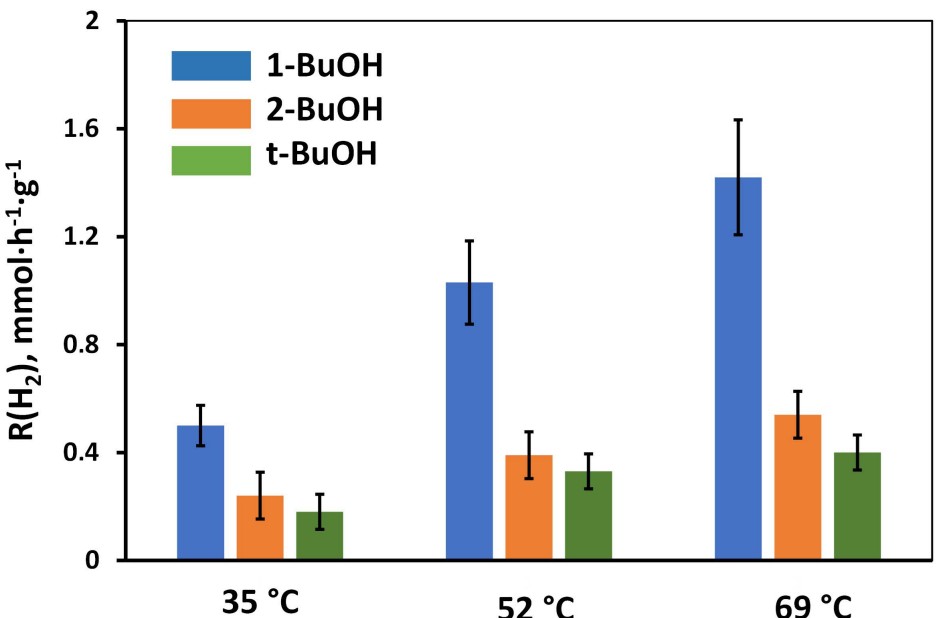

**Figure 3.** Rate of H$_2$ production as a function of bulk temperature from 0.5 M aqueous solutions of BuOH isomers irradiated with an Xe lamp (vis/NIR: 8.4 W, UV: 0.6 W) under Ar flow in the presence of Ti@TiO$_2$ NPs.

**Table 1.** The apparent activation energy of $H_2$ photocatalytic formation for BuOH isomers (0.5 M) irradiated with an Xe lamp in the presence of Ti@TiO$_2$ NPs under Ar flow.

| Alcohol | $E_a$, kJ·mol$^{-1}$ |
|---------|---------------------|
| 1-BuOH | $21 \pm 2$ |
| 2-BuOH | $20 \pm 2$ |
| t-BuOH | $13 \pm 2$ |
| Glycerol | $25 \pm 5$ * |

* Data [16].

In order to highlight the positive effect of temperature on $h^+$ mobility, we studied the photothermal production of $^•$OH radicals in Milli-Q water in the presence of Ti@TiO$_2$ NPs and without the addition of any alcohols. Recently, it was reported that $^•$OH radicals can be produced by oxidation of bridged O-H groups at the surface of TiO$_2$-based catalysts with the photogenerated $h^+$ [25]. Herein, the kinetics of $^•$OH radical formation was studied using terephthalate dosimetry [26]. This analytical technique is based on $^•$OH-induced hydroxylation of the terephthalate (TPH) ion and measurement of the formed 2-hydroxyterephtalate (2-HTPH) using fluorescence spectroscopy. The experimental details are described in Materials and Methods. Figure 4 clearly points out the acceleration of $^•$OH radical formation expressed as 2-HTPH concentration with the increase in bulk temperature in accordance with the above-discussed photothermal mechanism of $H_2$ production. The deceleration of 2-HTPH accumulation with irradiation time is most likely related to the photocatalytic oxidation of 2-HTPH.

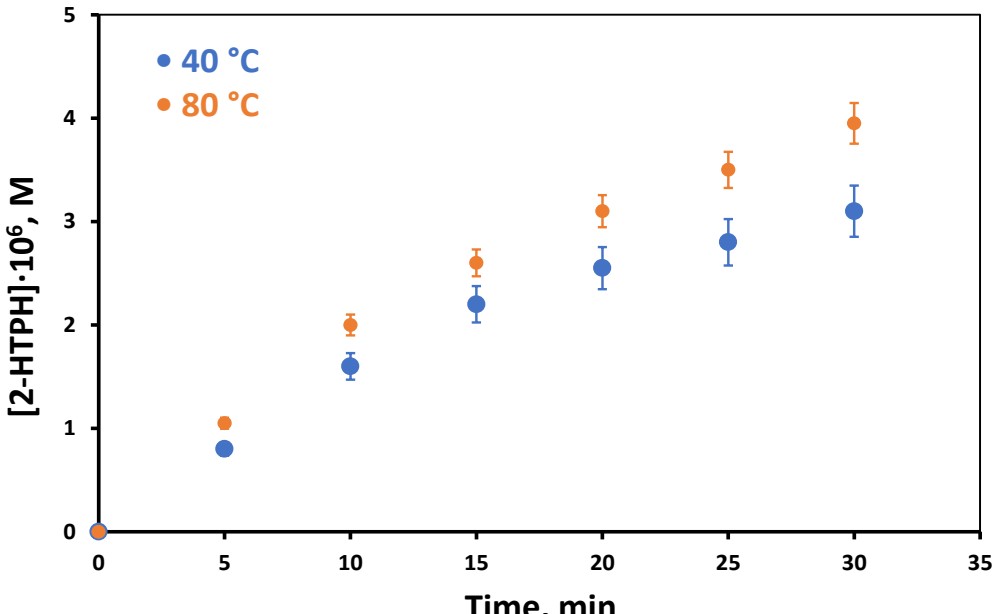

**Figure 4.** Kinetics of photocatalytic $^•$OH radical formation as a function of bulk temperature measured using terephthalate dosimetry in the presence of Ti@TiO$_2$ NPs and in the absence of butanol isomers (Xe lamp, Ar flow).

On the basis of the above results, it is evident that a rather low $E_a$ value for t-BuOH in comparison with 1-BuOH/2-BuOH isomers (Table 1) reflects the significant influence of C-C bond scission on the photothermal effect with t-BuOH. As aforementioned, the photothermal reforming of primary and secondary alcohols in the presence of TiO$_2$-based photocatalysts deals with the mobility of photogenerated $h^+$ [9,27,28]. In the system with t-BuOH, there are two possible reaction pathways for t-BuOH photoconversion: $h^+$-mediated O-H bond cleavage similar to that with 1-BuOH/2-BuOH isomers (Reactions 5,6), and

hydrogen abstraction from the C-H bond by the intermediate $OH^{\bullet}$ radical, leading to $CH_3^{\bullet}$ group ejection (Reactions 8,9) [16]:

$$[^{e^-}Ti@TiO_2{}^{h+}]\cdot(CH_3)_3COH \rightarrow [^{e^-}Ti@TiO_2]\cdot(CH_3)_3CO^{\bullet} + H^+ \tag{5}$$

$$[^{e^-}Ti@TiO_2] + H^+ \rightarrow H + Ti@TiO_2 \tag{6}$$

$$H + H \rightarrow H_2 \tag{7}$$

$$TiO_2\text{-}OH + h^+ \rightarrow TiO_2 + {}^{\bullet}OH \tag{8}$$

$$(CH_3)_3COH + {}^{\bullet}OH \rightarrow CH_2=C(CH_3)OH + CH_3^{\bullet} + H_2O \tag{9}$$

$$CH_3^{\bullet} + H^+ + e^- \rightarrow CH_4 \tag{10}$$

$$CH_3^{\bullet} + CH_3^{\bullet} \rightarrow C_2H_6 \tag{11}$$

The secondary reactions (10,11) leading to the emission of methane and ethane could influence the production of $H_2$. Actually, the $E_a$ measured experimentally involves not only the enthalpy of activation, $\Delta H^{\neq}$, but also the entropy of activation, $\Delta S^{\neq}$, which is sensitive to the reorganization of the solvent network during product formation and the migration of intermediates as well. In terms of the Eyring transition state theory, the free energy of activation, $\Delta G^{\neq}$, can be approximated as [29,30]:

$$E_a \approx \Delta G^{\neq} = \Delta H^{\neq} - T\Delta S^{\neq} \tag{12}$$

Hydrogen bonding in aqueous solutions induces the ordering of alcohol molecules at the surface of the catalyst. The ejection of the methyl group from t-BuOH would provide much stronger modification of the H–bonding network and, consequently, larger activation entropy than in the case of O–H bond cleavage for 1- and 2-BuOH isomers, which finally would lead to the drop of photothermal effect for t-BuOH.

### 3. Materials and Methods

#### 3.1. Chemical Reagents

1-BuOH, 2-BuOH, t-BuOH (all > 99%, Sigma-Aldrich, St. Louis, MI, USA), terephthalic acid, TPH (99%, Acros Organics, Waltham, MA, USA), 2-hydroxyterephthalic acid, 2-HTPH (97%, Sigma-Aldrich), NaOH (98%, Alfa Aesar, Haverill, MA, USA), $KH_2PO_4$ (99%, Sigma-Aldrich), and $Na_2HPO_4$ (99%, Sigma-Aldrich) were used as received without further purification. All solutions were prepared using Milli-Q grade ultrapure water (18.2 MΩ·cm at 25 °C). Pure Ar (99.999%) was supplied by Air Liquid.

#### 3.2. Catalyst Preparation

$Ti@TiO_2$ NPs were obtained by sonohydrothermal treatment (20 kHz, 200 °C) of commercially available $Ti^0$ NPs (American Elements) as described previously [19]. Typically, 2 g of air-passivated $Ti^0$ NPs was dispersed in 50 mL of Milli-Q water using an ultrasonic bath. The suspension was then transferred into the sonohydrothermal reactor and heated at 200 °C (autogenic pressure P = 19 bar) under simultaneous ultrasonic treatment (f = 20 kHz, $P_{ac}$ = 17 W) for 3 h. After cooling, the obtained particles were recovered using centrifugation, washed with deionized water, and dried at room temperature under reduced pressure.

#### 3.3. Photocatalytic Experiments

The photocatalytic study was performed in a thermostated glass-made gas-flow cell as shown in Figure 5. The temperature inside the reactor was controlled by an external thermostat. In a typical run, 7.8 mg of photocatalyst was dispersed ultrasonically in 65 mL of an aqueous solution of alcohol before transferring it into the photoreactor. The cell used for the photocatalytic experiments was equipped with two inlets, one to purge the gas and another to measure the temperature during photothermal treatment. The argon gas

flow through the reactor was controlled by a volumetric flowmeter and kept constant at $45~\text{mL·min}^{-1}$ during the experiment. Photocatalytic experiments were carried out using the white light of an ASB-XE-175W xenon lamp. During the experiments, the lamp was placed at a distance of 8 cm away from the reactor. The light intensity delivered onto the reactor at such distance was measured with an X1-1 Optometer (Gigahertz-Optik) using UV-3710-4 (300–420 nm) and RW-3705-4 (400–1100 nm) calibrated detectors. The obtained values were normalized to the irradiated surface area, and the calculated light power was equal to 8.4 W and 0.6 W for vis/NIR and UV spectral ranges, respectively. During the photocatalytic treatment, the suspensions inside the reactor were stirred continuously, and the temperature was increased stepwise up to 69 °C. The cell used for performing the photocatalytic experiments was also equipped with an outlet connected to a bench-top magnetic sector mass spectrometer (Themo Scientific PRIMA BT, Waltham, MA, USA), allowing for the continuous online analysis of the produced gases. The $H_2$ formation was quantified using external calibration curves prepared with standard gas mixtures in argon (Air Liquide).

**(A)**                  **(B)**

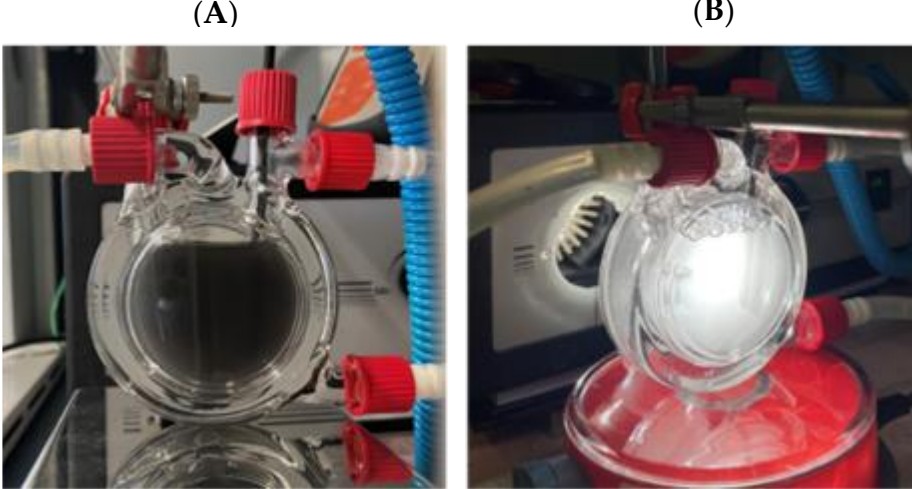

**Figure 5.** Image of the photocatalytic cell before (**A**) and during (**B**) illumination with xenon lamp.

### 3.4. Terephthalate Dosimetry

The solution for terephthalate dosimetry was prepared as follows: 332 mg of terephthalic acid was dissolved under mechanical stirring at an ambient temperature in a 500 mL buffer solution prepared with 200 mg of NaOH, 590 mg of $KH_2PO_4$, and 980 mg of $Na_2HPO_4$ dissolved in Milli-Q water. The obtained solution was then diluted to 1 L with water. Then, 7.8 mg of photocatalyst was dispersed into 65 mL of the prepared solution using an ultrasonic bath and further introduced into the photocatalytic cell. During photolysis, sample aliquots were withdrawn from the reactor every 5 min and filtered through 0.2 μm PTFE filters. The concentration of 2-hydroxyterephthalic acid formed upon the reaction of terephthalic acid with $^\bullet$OH radicals was measured using fluorescence at 425 nm using a Fluoromax-4 device equipped with a Horiba NanoLED laser providing an excitation wavelength of 315 nm. A calibration curve has been obtained using standard solutions of 2-hydroxyterephthalic acid.

### 4. Conclusions

The photothermal hydrogen production from the aqueous solutions of 1-BuOH, 2-BuOH, and t-BuOH isomers studied in this work provides new insights into reaction mechanisms. The experiments were performed using innovative, noble, metal-free core-shell Ti@TiO$_2$ nanoparticles obtained through hydrothermal oxidation of metallic titanium nanoparticles assisted by power ultrasound. We found that the isomerization of butanol strongly influences the kinetics of photocatalytic hydrogen evolution and its thermal response. The 1-BuOH isomer exhibits the highest $H_2$ formation rate, which corroborates

with the strongest adsorption of 1-BuOH at the surface of $TiO_2$ compared to other butanol isomers. Analysis of the gaseous products revealed a significant difference between 1-BuOH/2-BuOH and t-BuOH. Photolysis of 1-BuOH/2-BuOH solutions yields solely $H_2$ as a gaseous photocatalytic product without emission of $CO_2$ or hydrocarbons. However, the photocatalytic degradation of t-BuOH leads to the formation of $H_2$, $CH_4$, and $C_2H_6$ indicating the scission of the C–C bond for tertiary isomers. A similar trend is observed in the thermal response of the photocatalytic $H_2$ formation. The apparent activation energies, $E_a$, are very similar for 1-BuOH and 2-BuOH isomers (20–21 kJ·mol$^{-1}$), and are also fairly close to that of glycerol measured in our earlier studies with the same catalyst. On the other hand, the $E_a$ value for t-BuOH is much lower (13 kJ·mol$^{-1}$), indicating a weak photothermal effect for this isomer. This difference has been attributed to the more complex mechanism in the case of tertiary isomers involving a large number of intermediates and the contribution of photogenerated $^\bullet$OH radicals to the degradation of t-BuOH. Formation of $^\bullet$OH radicals during photoexcitation of Ti@$TiO_2$ nanoparticles was confirmed using terephtalate dosimetry.

**Author Contributions:** Conceptualization, S.I.N.; methodology, S.E.H. and T.C.; formal analysis, S.I.N., S.E.H. and T.C.; investigation, S.E.H. and M.B.; data curation, S.E.H. and T.C.; writing—original draft preparation, S.E.H. and M.B.; writing—review and editing, S.I.N. and S.E.H.; visualization, S.E.H. and S.I.N.; supervision, S.I.N. All authors have read and agreed to the published version of the manuscript.

**Funding:** This research received no external funding.

**Data Availability Statement:** The data presented in this study are available in the article.

**Conflicts of Interest:** The authors declare no conflict of interest.

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
