# Peer review of "Influence of Butanol Isomerization on Photothermal Hydrogen Production over Ti@TiO2 Core-Shell Nanoparticles"

_catalysts, doi:10.3390/catal12121662_

Round 1
Reviewer 1 Report
Dear authors
The manuscript entitled “Influence of butanol isomerization on photothermal hydrogen 2 production over Ti@TiO2 core-shell nanoparticles” has an interesting topic for readers of Catalyst journal. The manuscript needs major revision and the comments for increasing the quality of the manuscript are presented below.
- The symbol of the reactivity state of hydroxyl radical should be inserted on the oxygen atom. (lOH).
- What was the bandgap of the Ti@TiO2 catalyst?
- If the light is used (photocatalytic reaction), the DRS test should be done to determine the bandgap and the maximum absorption wavelength.
- What was the proportion of heat, ultrasonic, light, and catalyst in the hydrogen production process?
- There are several reactive radicals in photocatalytic reactions and their domain of them can be determined using scavenging tests.
- The experiments were done in pure water. It is recommended to do a test in an optimized situation in tap water. Due to the scavenging role of anions in tap water, it was proposed to optimize the results in real waters.
Author Response
1. The symbol of the reactivity state of hydroxyl radical should be inserted on the oxygen atom.
Answer: The symbol of radical has been placed on the oxygen atom throughout the text.
2. What was the bandgap of the Ti@TiO2 catalyst? If the light is used (photocatalytic reaction), the DRS test should be done to determine the bandgap and the maximum absorption wavelength.
Answer: In this work, we used the Ti@TiO2 catalyst characterized in our previous papers (see Refs. 15-17 of the revised paper). The bandgap energy value for nanocrystalline TiO2 shell obtained by DRS technique was found to be equal to 3.52±0.1 eV (Ref. 17 of the revised paper). This value has been added to the revised paper (L65).
3. What was the proportion of heat, ultrasonic, light, and catalyst in the hydrogen production process? There are several reactive radicals in photocatalytic reactions and their domain of them can be determined using scavenging tests.
Answer: Ultrasound in combination with hydrothermal conditions was used for the synthesis of catalyst and not for the photocatalytic hydrogen production. Therefore, there is no direct contribution of ultrasound in the process of hydrogen production. On the other hand, our work shows that the introducing heat into the photocatalytic process largely enhances the production of hydrogen with studied butanol isomers. For example, increase of the bulk temperature from 35 °C to 69 °C in the system with 1-BuOH leads to 3 times increase of hydrogen production rate for the similar other conditions (Figure 3).However, Figure 2 points out that hydrogen is not formed at dark conditions even at elevated temperature. Therefore, light absorption by the catalyst is a primary parameter in studied photothermal process. Regarding the formation of intermediate radicals, we provided the evidence of OH radical formation with the studied catalyst using terephtalic acid as a scavenger. These results are shown in Figure 4. In the literature (Ref. 11 of the revised paper), the formation of CH3 radicals was observed experimentally during the UV-driven photolysis of t-BuOH over TiO2 in the gas phase. In our work, using of radical scavengers in aqueous solutions of butanol in the presence of photocatalyst could lead to several secondary photocatalytic processes and the results will be hard to interpret. In the system with t-BuOH, formation of CH4 and C2H6 clearly points out on the formation of intermediate CH3 radicals. In the revised paper, a following modification was done at the page 4, L172-175 :
“However, for the system with t-BuOH the formation of CH4 and C2H6 was detected at 35-52 °C along with H2 formation, which agrees with the observation of CH3• radicals during photolysis of t-BuOH in the gas phase over TiO2 [12].”
4. The experiments were done in pure water. It is recommended to do a test in an optimized situation in tap water. Due to the scavenging role of anions in tap water, it was proposed to optimize the results in real waters.
Answer: We do agree with the reviewer that the anions and cations presenting in tap water would influence the rate of photocatalytic processes. It could be a topic of future study since this work focuses on the influence of butanol isomerization on the process of photothermal hydrogen production. Regarding future large-scale applications of the photothermal catalysis, the aqueous solutions can be prepared with demineralized water obtained by the photothermal evaporation of tap or even sea water in the presence of sunlight adsorbing materials, such as Ti@TiO2 particles, using middle-power solar concentrators.
Reviewer 2 Report
Comment 1: The abstract is ok. However, the author should highlight the novelty in the abstract.
Comment 2: The introduction is very limited and poor. The author must elaborate and add more literature in it to make the article up to the standard of ‘catalysts’. The novelty statement and the significance of the work should be clearly stated as a separate and last paragraph of the introduction section.
Comment 3: The introduction is poor in the literature review. The author may add the following suggested references to make the literature richer i.e., https://doi.org/10.1016/B978-0-12-823007-7.00015-8 , https://doi.org/10.1016/j.rser.2022.112916 and https://doi.org/10.1016/j.ces.2020.116072
Comment 4:. Some of the statements are too long resulting in low clarity. The manuscript needs to be proofread for better clarity and to eliminate grammatical errors.
Comment 5: The author may add the chart/figure to show the “percentage efficiency” of the hydrogen production varying with temperature since it is an important parameter.
Comment 6: In Section “3.2. Catalyst Preparation” Lines 259-261 “heated at 200 °C (autogenic pressure P = 19 bar) under simultaneous ultrasonic treatment (f = 20 kHz, Pac = 17 W) for 3 h. After cooling, the obtained particles were recovered by centrifugation, washed with deionized water, and dried at room temperature under reduced pressure”
Did the author adopt methodology from any literature? If yes, the author must cite the relevant literature. If not justify the reason of heating at 200oC and ultrasonication for 3h.
Comment 8: The conclusion resembles almost the same as the abstract. A detailed conclusion summarizing the results (and concluding statements from each subsection of the experiment) need to be added at the end of the manuscript.
Comment 9: There are very few references/studies cited to support the results and discussion. Apart from the above-suggested references, the author must add more studies to support the literature review in the introduction, methodology, and results & discussion sections.
Author Response
Comment 1: The abstract is ok. However, the author should highlight the novelty in the abstract.
Answer: In order to highlight the major finding of this work following modifications have been made in the Abstract:
“In this work, we reported for the first time the effect of butanol isomerization on the photothermal production of hydrogen in the presence of noble metal-free Ti@TiO2 core-shell photocatalyst.”
“Formation of •OH radicals during light irradiation of Ti@TiO2 nanoparticles suspension in water has been confirmed using terephthalate dosimetry. This analysis also revealed a positive temperature response of •OH radicals formation.”
Comment 2: The introduction is very limited and poor. The author must elaborate and add more literature in it to make the article up to the standard of ‘catalysts’. The novelty statement and the significance of the work should be clearly stated as a separate and last paragraph of the introduction section.
Comment 3: The introduction is poor in the literature review. The author may add the following suggested references to make the literature richer i.e., https://doi.org/10.1016/B978-0-12-823007-7.00015-8, https://doi.org/10.1016/j.rser.2022.112916 , and https://doi.org/10.1016/j.ces.2020.116072.
Answer to the comments 2 and 3: Thank you for suggestions. In the Introduction of revised paper, we cited the works recommended by the reviewer and some other papers relevant to green hydrogen production (Ref. 1-4).
Comment 4:. Some of the statements are too long resulting in low clarity. The manuscript needs to be proofread for better clarity and to eliminate grammatical errors.
Answer: The manuscript has been proofread to improve its readability.
Comment 5: The author may add the chart/figure to show the “percentage efficiency” of the hydrogen production varying with temperature since it is an important parameter.
Answer: We agree with the reviewer that the variation of hydrogen production efficiency with temperature for different butanol isomers in an important parameter for this work. Actually, such a comparison is already demonstrated in Fig. 3, where the hydrogen production rate at different temperatures is compared for three butanol isomers. More quantitatively, the thermal effect of the studied photocatalytic process is expressed in terms of apparent activation energy, Ea (Table 1). The established relationship between the Ea values and butanol isomers provided new insights into the origin of the photothermal effect discussed in this paper.
Comment 6: In Section “3.2. Catalyst Preparation” Lines 259-261 “heated at 200 °C (autogenic pressure P = 19 bar) under simultaneous ultrasonic treatment (f = 20 kHz, Pac = 17 W) for 3 h. After cooling, the obtained particles were recovered by centrifugation, washed with deionized water, and dried at room temperature under reduced pressure”
Did the author adopt methodology from any literature? If yes, the author must cite the relevant literature. If not justify the reason of heating at 200oC and ultrasonication for 3h.
Answer: Yes, the catalyst was prepared by the innovative methodology developed recently by our group and published in our earlier works. The relevant reference describing the optimization of the synthesis was inserted into the revised paper.
Comment 8: The conclusion resembles almost the same as the abstract. A detailed conclusion summarizing the results (and concluding statements from each subsection of the experiment) need to be added at the end of the manuscript.
Answer: We agree with this comment. In the revised paper, the fundamentally revised Conclusions are presented:
"The photothermal hydrogen production from the aqueous solutions of 1-BuOH, 2-BuOH, and t-BuOH isomers studied in this work provided new insights upon reaction mechanisms. The experiments were performed using innovative noble metal-free core-shell Ti@TiO2 nanoparticles obtained by hydrothermal oxidation of metallic titanium nanoparticles assisted by power ultrasound. We found that the isomerization of butanol strongly influences the kinetics of photocatalytic hydrogen evolution and its thermal response as well. The 1-BuOH isomer exhibits the highest H2 formation rate, which corroborates with the strongest adsorption of 1-BuOH at the surface of TiO2 compared to other butanol isomers. Analysis of the gaseous products revealed significant difference between 1-BuOH/2-BuOH and t-BuOH. Photolysis of 1-BuOH/2-BuOH solutions yields solely H2 as a gaseous photocatalytic product without emission of CO2 or hydrocarbons. However, the photocatalytic degradation of t-BuOH leads to the formation of H2, CH4 and C2H6 indicating the scission of C-C bond for tertiary isomer. Similar trend is observed in the thermal response of the photocatalytic H2 formation. The apparent activation energies, Ea, are very similar for 1-BuOH and 2-BuOH isomers (20-21 kJ·mol-1) and also fairy close to that of glycerol measured in our earlier studies with the same catalyst. On the other hand, the Ea value for t-BuOH is much lower (13 kJ·mol-1) indicating weak photothermal effect for this isomer. Such a difference has been attributed to a more complex mechanism in the case of tertiary isomer involving large number of intermediates and the contribution of photogenerated •OH radical into the degradation of t-BuOH. Formation of •OH radicals during photoexcitation of Ti@TiO2 nanoparticles was confirmed using terephtalate dosimetry."
Comment 9: There are very few references/studies cited to support the results and discussion. Apart from the above-suggested references, the author must add more studies to support the literature review in the introduction, methodology, and results & discussion sections.
Answer: As it was mentioned in our answer to the comments 2 and 3, we have added several relevant references to the revised paper. Meanwhile, despite a huge number of publications on heterogeneous photocatalytic hydrogen production only few of them focus on the comparative study of butanol isomers during photodegradation. We hope that we have used the most appropriate references for this work.
Round 2
Reviewer 2 Report
The author incorporated the suggestions successfully, which increased the overall quality of the manuscript. The article maybe considered for acceptance. However, the literature review can be improved further by adding the a few more relevant references as suggested in the revision.
Author Response
According to the reviewer 2 comment, we added the references 4, 13, and 30 to the revised paper.